



# Modern Pollen Dataset of the Tibetan Plateau

Mengna Liao[1], Kai Li[1], Yili Jin[1], Lina Liu[1], Xianyong Cao[2], Jian Ni[1]

[1]College of Life Sciences, Zhejiang Normal University, Jinhua 321004, China
[2]Key Laboratory of Alpine Ecology, CAS Center for Excellence in Tibetan Plateau Earth Sciences, Institute of Tibetan
Plateau Research, Chinese Academy of Sciences, 100101 Beijing, China

*Correspondence to*: Jian Ni (nijian@zjnu.edu.cn)

**Abstract.** Modern pollen datasets can provide invaluable data for interpreting temporal variations in climate, vegetation, land cover, and plant diversity from fossil pollen. Here we present 555 pollen count data, identified from topsoil collected within plant plots across a vast area of the Tibetan Plateau (TP) and along the southern margin of Xinjiang that borders the
TP. This dataset fills a geographical gap in the published datasets that offer pollen count data for this area. Ordination analysis and multiple regression reveal that precipitation is the primary factor influencing the spatial distribution of pollen assemblages across the entire study area. Furthermore, ordination analysis indicates that pollen assemblages can be used to distinguish vegetation types in the southeastern TP, such as coniferous forest, alpine shrubland, and alpine meadow, from vegetation types found in other regions of TP. Additionally, it is possible to distinguish vegetation types that have low
precipitation or moisture requirements based on pollen assemblages. Generalized additive models demonstrate that six commonly used pollen ratios, involving taxa such as *Artemisia*, Amaranthaceae, Cyperaceae, and Poaceae, are not sufficiently reliable for reflecting changes in annual precipitation. Nevertheless, they can provide some indication of changes in vegetation or landscape. This dataset holds various potential applications in paleoecological and paleoclimatic researches. It not only offers a scientific foundation for reconstructing changes in climate and vegetation over time, but also enables the
assessment of the reliability of pollen assemblages in representing the dynamics of vegetation cover, functional traits, and plant diversity, by integrating the simultaneously measured plot-level plant communities and functional traits. Data from this study, including pollen count data for each sample and site, alongside with the geographical coordinates, altitude, local vegetation type of each sampling site, dry weight of each sample used for pollen extraction, Lycopodium (marker) grains per tablet, and the identified number of Lycopodium spores, are available at https://doi.org/10.11888/Paleoenv.tpdc.302015
(Liao et al., 2025).

## 1 Introduction

Modern pollen samples, cross-referenced with current distributions of climate and vegetation, provide invaluable data for interpreting temporal variations in climate, terrestrial vegetation, and land cover from fossil pollen data (Prentice et al., 1996; Zhu et al., 2010; Fyfe et al., 2015; Kaufman and Broadman, 2023; Liu et al., 2024). Over the past four decades, modern
pollen datasets have been progressively established at continental or subcontinental scales, such as the European Modern



Pollen Database (Davis et al., 2013), Eurasian Modern Pollen Database (Davis et al., 2020), East Asian Pollen Database (Zheng et al., 2014), Modern Pollen Dataset of China (Chen et al., 2021), modern pollen data from North America and Greenland (Whitmore et al., 2005), and Latin American pollen database (Flantua et al., 2015). Public availability of these modern pollen data can facilitate quantitative reconstructions of past climate, biomes, land cover, and plant diversity, and

additionally aid in creating benchmarks for evaluation vegetation and climate simulations.

Tibetan Plateau (TP), renowned as the world's "Third Pole" and the cradle of East Asian flora, constitutes the largest area of uplifted crust on Earth. It plays a pivotal role in the formation of the climate pattern and hydrological system in East Asia (Yao et al., 2012), as well as in the evolution of flora, fauna and biodiversity (Ding et al., 2020). Besides, TP is highly sensitive to global climate change, and its landscapes are fragile (Chen et al., 2015; Ehlers et al., 2022). Due to its unique

environment, TP has been a hotspot for studying present and past changes in climate and vegetation (Herzschuh et al., 2009; Li et al., 2022; Zhou et al., 2024), and for investigating interactions between human and environmental change (Gao et al., 2022; Zhang et al., 2022). Over the past two decades, modern pollen datasets from TP (Yu et al., 2001; Herzschuh et al., 2010; Lu et al., 2011; Cao et al., 2014; Cao et al., 2021; Wang et al., 2022; Ma et al., 2024b) or those from China or the world that include a substantial number of modern pollen samples collected from TP have been published (Zheng et al., 2014;

Davis et al., 2020; Chen et al., 2021; Cui et al., 2024). Some of these datasets are publicly accessible, including lake surface sediment pollen datasets on the eastern, central, and western TP (Cao et al., 2021; Ma et al., 2024b), the Modern Pollen Dataset of China (Chen et al., 2021), and the Eurasian Modern Pollen Database (Davis et al., 2020). However, most of these data have either been digitized or converted into percentages, and collected from multiple sources, making it difficult to evaluate their quality and potentially introduce additional bias or uncertainties into further analysis.

Here we present a modern pollen count data identified by the team themselves from 555 topsoil samples collected within plant plots on TP and in the southern margin of Xinjiang bordering with the northern plateau. We aim to: (i) fill a geographical gap left by previous datasets that provided pollen count data from the TP, and (ii) propose potential uses of this dataset and necessary considerations for its application. This modern pollen dataset not only provides a probability to comprehensively dissect the linkages between pollen assemblage and climatic variables, as well as vegetation, across large

gradients of climate and vegetation, but also help improving the accuracy of reconstructions of regional climate, vegetation, and land cover.

## 2 Study area

The study area covers a wide geographical range between 28–40°N latitude and 75–103°E longitude, including large areas of the TP and the southern margin of Xinjiang, which borders the northern plateau (Fig. 1). Due to the synthetic effects of

various factors such as altitude, topography, and atmospheric circulation, TP exhibits a prominent gradient distribution in climate (China Meteorological Data Service Centre, https://data.cma.cn). The TP is strongly limited by thermal deficiency, with mean annual temperature (MAT) across the surface plateau almost below 0℃. The MAT decreases from eastern to



western TP, in which the MAT on the northern and eastern parts of the TP ranges from 0 to 10°C but decrease to about -

10°C in western TP. The precipitation generally decreases from southeast impacted by Indian Summer Monsoon to

northwest influenced by dry westerly. The mean annual precipitation (MAP) reaching several hundred to even over a

thousand millimeters on the southeastern TP, while decreases to below 50 mm on the northwestern TP. Additionally,

precipitation on TP exhibits distinct seasonal variations, with the majority of the annual precipitation occurring during the

summer. Due to the pronounced climate gradients, the vegetation on the plateau transitions successively from montane forest

in the southeast to alpine shrub and meadow, followed by alpine steppe in the middle, and finally alpine desert in the

northwest. In the northeastern TP and the southern margin of Xinjiang, where the altitude is relatively low, the vegetation is

predominantly temperate and consists primarily of meadow, steppe, and desert.

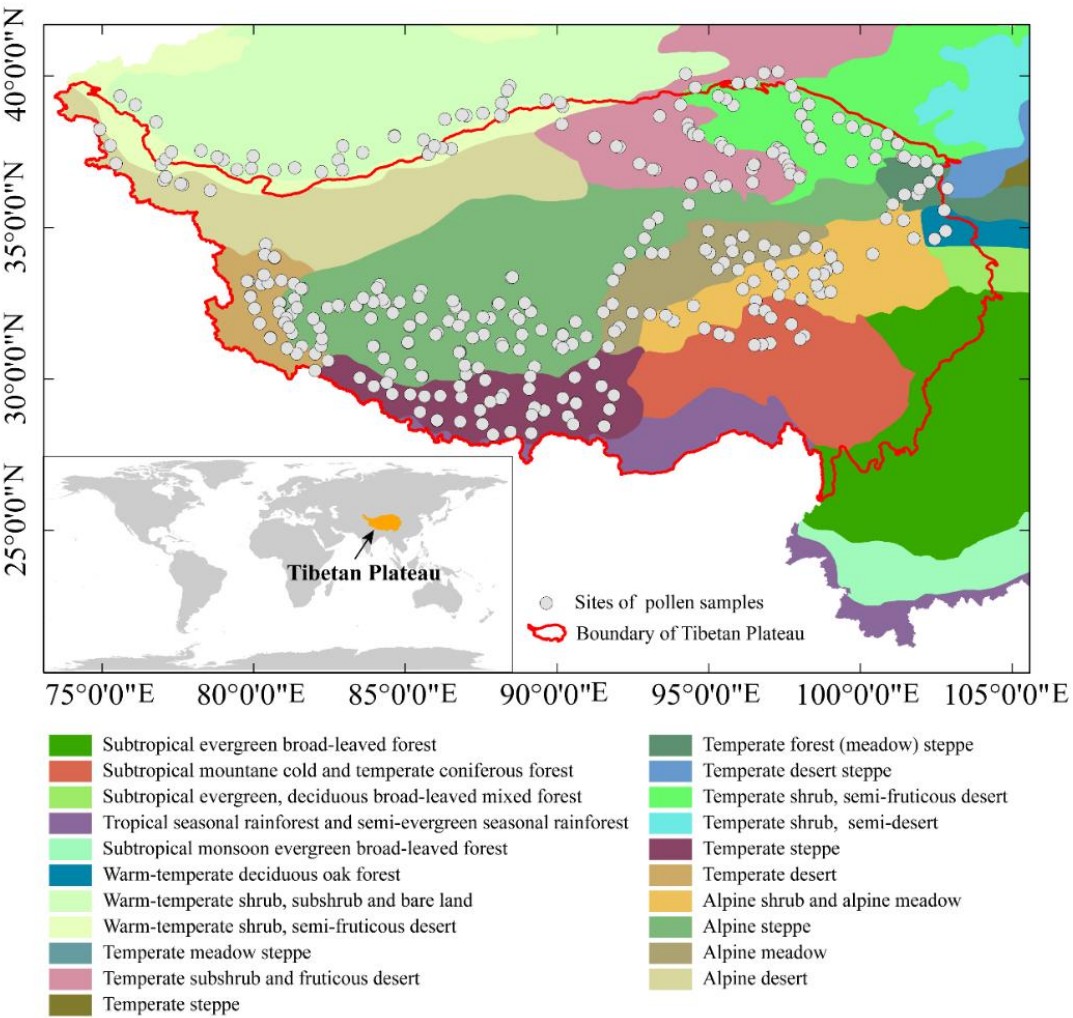

**Figure 1: Geological locations of the Tibetan Plateau (TP) and the sampling sites. The background map showing the distribution of modern vegetation across the TP and its surrounding regions in China (Zhang, 2007).**





## 3 Materials and methods

### 3.1 Sample collection


A total of 664 topsoil samples were collected from 307 sites across the study area between 2018 and 2022 (Fig. 1). The sampling sites cover diverse vegetation types and span a broad altitude range. The sample sites in alpine vegetation are generally located above 3000 m a.s.l., while those in temperate vegetation distribute between 800 and 3000 m a.s.l. (Jin et al.,

2022). Based on the local vegetation surrounding the sampling sites (Jin et al., 2022), the samples can be classified into eight groups: coniferous forest (3 samples from 3 sites), alpine shrubland (54 samples from 35 sites), alpine meadow (136 samples from 80 sites), alpine steppe (191 samples from 89 sites), alpine desert (38 samples from 15 sites), temperate meadow (21 samples from 15 sites), temperate steppe (18 samples from 9 sites), and temperate desert (94 samples from 61 sites). It should be noted that the geographical coverage of the samples collected in 2018 for this dataset overlaps with that of the

dataset published by Cao et al. (2021). However, the geographical locations of the samples in the two datasets are different. Additionally, the samples collected by our team are topsoil samples, whereas those collected by Cao et al. (2021) are from lake surface sediments.

### 3.2 Pollen analysis

The samples were pretreated using the heavy liquid floatation method (Moore et al., 1991; Nakagawa et al., 1998) involving

five main steps: removing carbonate with 10% HCl, removing humic substances with 10% KOH, sieving through a 125 μm mesh to remove gravels and plant roots, performing heavy liquid flotation with a zinc bromide solution (ZnBr2 at approximately 2.2 g ml-1), and removing cellulose with acetolysis. *Lycopodium* spores (27,560 grains per tablet for samples of 2018, and 10,315 grains per tablet for samples of 2019, 2020 and 2021) were added to each sample prior to the pretreatment. Pollen identification referred to the Chinese pollen books (Wang et al., 1995; Tang et al., 2016). It is noted that

for samples collected in 2018, those collected from different plots within the same site were mixed evenly before been used for pollen extracting. Ultimately, 555 pretreated samples were used for pollen identification. In the dataset, sample identities that differ only in their suffix letters indicate that they were collected from different quadrats within the same site.

### 3.3 Numerical analyses

Pollen percentages were determined by calculating the proportion of each pollen type relative to the total number of pollen

grains identified in each sample, and these percentages were subsequently utilized for numerical analyses. To explore the similarities among pollen assemblages, we performed non-metric multidimensional scaling (NMDS) using Bray-Curtis distance as the distance measure between different pollen assemblages. Additionally, to assess the relations of bioclimatic variables to the first two ordination axes of the NMDS, we conducted multiple regression analysis with bioclimatic variables as dependent and the ordination axes as explanatory variables. The significance of the relations was tested using a

permutation test with 999 permutations. We used eight bioclimatic variables extracted from a 1-km resolution dataset on

climate changes over China (Hu et al., 2024), including MAT ($T_{ann}$), temperature seasonality ($T_{season}$), mean temperature of the warmest quarter ($T_{warm}$), mean temperature of the coldest quarter ($T_{cold}$), MAP ($P_{ann}$), precipitation seasonality ($P_{season}$), precipitation of the wettest quarter ($P_{wet}$), and precipitation of the driest quarter ($P_{dry}$). These analyses were conducted using the R package "vegan" (Oksanen et al., 2022).

Six commonly used pollen ratios, including Artemisia/Amaranthaceae (A/Am), Artemisia/Cyperaceae (A/Cy), Poaceae/*Artemisia* (Po/A), Cyperaceae/Poaceae (Cy/Po), Cyperaceae/(Poaceae+*Artemisia*) (Cy/(Po+A)), and Poaceae/(*Artemisia*+Amaranthaceae) (Po/(A+Am)), were selected to investigate their relationships with MAP. We applied generalized additive models (GAMs) to uncover these relationships, as GAMs are nonparametric data-driven regression models that can effectively assess nonlinear relationships between response and predictor variables without any restrictive

assumptions (Hastie and Tibshirani, 1987). The significance of these relationships was tested using F-tests, with a significance level $\leqslant 0.05$ considered significant. The R package "mgcv" (Wood, 2011) was used for performing the GAMs.

## 4 Data description

### 4.1 Sample collection

The number of identified pollen grains in this dataset varies greatly, with a minimum of 6 grains per sample (or 10 grains per

site) and a maximum of 1331 grains per sample (or 2626 grains per site). Samples or sites with relatively low pollen count are mainly from some plots on hillside distributing in the southwestern and northeastern TP (Fig. 2). Therefore, we emphasize the importance of conducting data filtering before using this dataset. According to the frequency distributions of the pollen counts, samples with counts between 200 and 300 constitute a relatively large proportion, accounting for approximately 31% (Fig. 2). This is followed by the number of samples with counts between 500 and 600, and between 400

and 500. When considering the total counts at each sampling site, 44% of the sites have counts ranging between 400 and 600 (Fig. 2).





**Figure 2: Spatial distributions of the number of pollen grains identified from each sample (upper panel) and from each site (lower panel). The inserted histograms showing the frequency distributions of the pollen count.**


A total of 145 identified pollen types have been recorded in the dataset. Based on the distribution of median values across different vegetation types, samples collected from coniferous forest, alpine shrubland, alpine meadow, and temperate steppe contain a relatively large number of pollen taxa (Fig. 3). When considering the total number of identified pollen types at each sampling site, sites in temperate steppe generally contain a distinctly larger number of pollen taxa compared to those in other

vegetation types, whereas sites in temperate meadow and desert contain the fewest pollen taxa (Fig. 3). It should be noted that, since the pollen types of the samples collected each year from 2018 to 2021 were identified by different individuals, there may exist slight difference in the taxonomic resolution among these pollen data. For example, *Abies* and *Picea* were not distinguished in the samples collected in 2018, but they were separated in the samples collected in 2019, 2020, and 2021. Given that this dataset may be used by different scientists who may have different requirements for data processing, we

decided to preserve the aboriginality of the data in the dataset, i.e. not to homogenize the taxonomy at the family or genus level, and to retain the counts for unidentified pollen type(s) (marked as "Unkown" in the dataset).



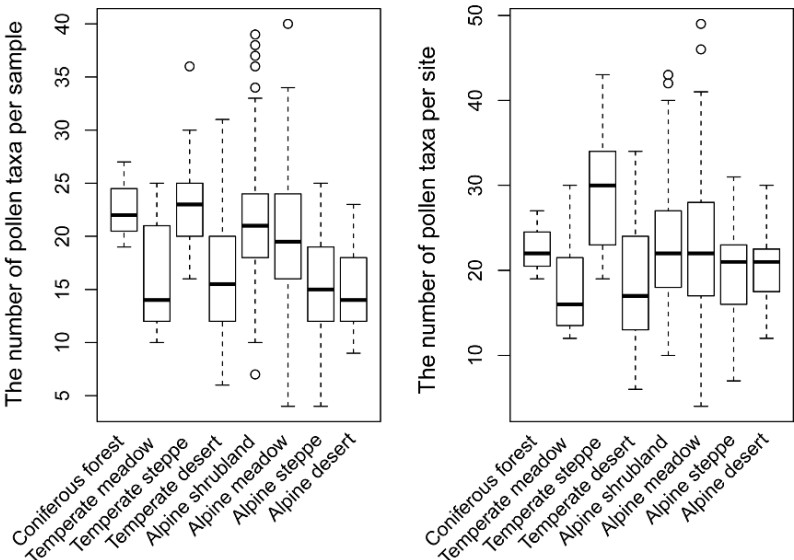

**Figure 3: Boxplots showing the distribution of the number of identified pollen taxa from different vegetation types**

### 4.2 Distributions of major pollen taxa


The percentage abundance of the main pollen types exhibits different spatial distribution patterns (Fig. 4). *Pinus* reaches a maximum percentage of approximately 47%, with percentages between 20–30% and between 30–50% primarily observed in the northeastern and southwestern TP, respectively. *Ephedra* shows a relatively high percentage (>30%) in the central and eastern parts of the southern margin of Xinjiang. Amaranthaceae displays relatively high percentages primarily in the

northeastern and southwestern TP, as well as along the southern margin of Xinjiang. *Artemisa* is relatively abundant in the northeastern, northwestern, and southern regions of TP, and in the central part of the southern margin of Xinjiang. Cyperaceae exhibits a distinct special pattern, with high percentages observed in the eastern, southeastern, and southern areas of TP. Ranunculaceae has notable high percentages at a few sites along the southern margin of Xinjiang. A few sampling sites with relatively high percentages of *Tamarix* are located in the Qaidam Basin, a relatively low-altitude area in

the northeastern TP. For the remaining pollen taxa, the majority of the percentages are below 10%, and there is no obvious difference in their spatial distributions.





**Figure 4: Spatial distributions of seventeen major pollen taxa (each with maximum percentage ≥ 10% and simultaneously included at least three samples with a percentage of ≥10%). The background map showing the distribution of MAP (mm) across the study area (Hu et al., 2024).**





## 5 Potential usage of the dataset

The modern pollen dataset of TP has several potential uses, particularly in assessing the reliability of the pollen assemblages in reflecting changes in climate and vegetation (in terms of composition, vegetation coverage, functional trait, and biodiversity). According to the results of NMDS and multiple regression, all the selected climatic variables exhibit significant correlations ($p < 0.001$) with the first two axes of the MNDS. It suggests, as has been demonstrated by other studies (Lu et al., 2011; Ma et al., 2024a), that pollen assemblages have the potential to provide reliable estimates of climatic parameters on TP. As indicated by the $R^2$ values, $P_{ann}$ and $P_{wet}$ are the most promising in reflecting changes in climate across the entire study area, followed by $T_{warm}$, and then $P_{season}$ and $T_{ann}$ (Fig. 5). Pollen ratios have commonly been used to indicate changes in landscape and climate. GAMs reveal that the relationships of pollen ratios (except for Po/A) with MAP exhibits significant but (extremely) weak correlations for each pair (Fig. 6). This demonstrates that, at a geographical scale across the entire study area, none of these ratios are reliable indicators of precipitation changes (Fig. 6). However, when comparing these ratios across different vegetation types, it is evident that they can reflect changes in vegetation or landscape to some extent (Fig. 6). Specially, the A/Am ratio shows relatively high values in alpine shrubland, alpine desert, and alpine steppe. The A/Cy ratio in alpine steppe, temperate steppe, and temperate desert displays distinctly high values compared to that in other vegetation types. The Po/A ratio and Po/(A+Am) ratio of samples from temperate steppe and temperate desert are notably low compared to samples from other vegetation types. The Cy/Po ratio and Cy/(Po+A) ratio reflect a similar pattern across vegetation types, showing relatively high values in alpine shrubland and alpine desert.

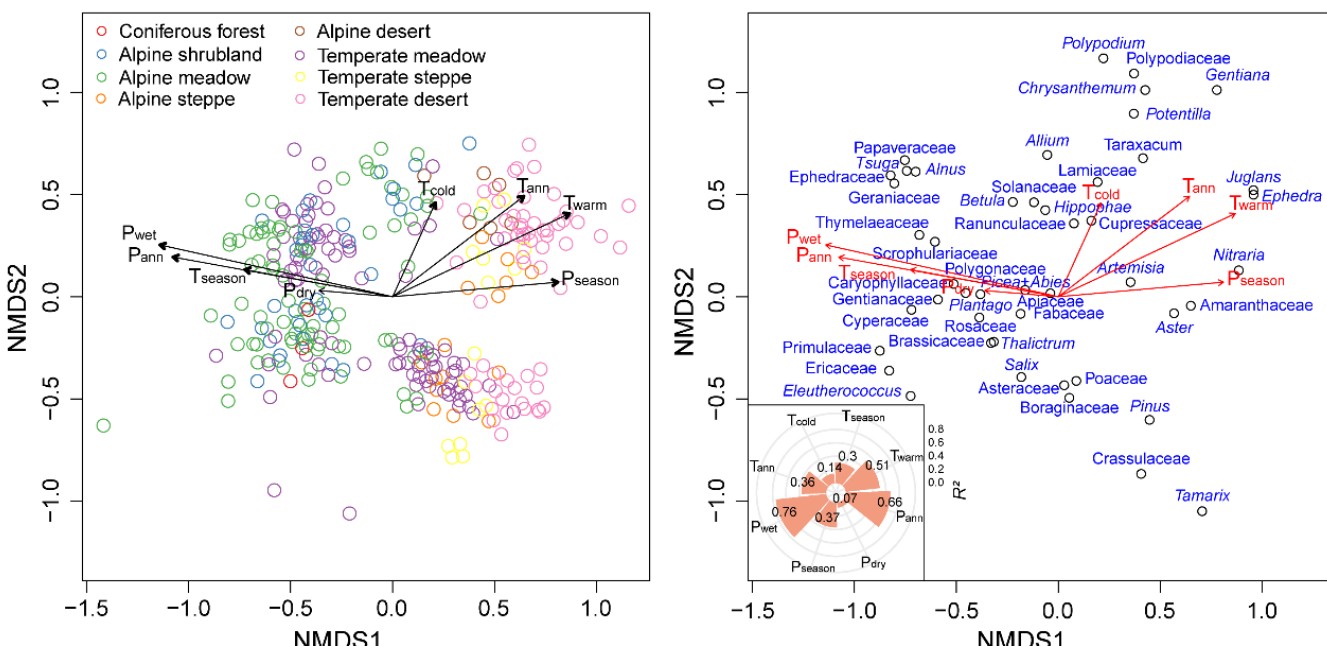

**Figure 5: Biplots of non-metric multidimensional scaling (NMDS) showing the sites (left) and the main pollen taxa (right). The inserted polar plot showing the coefficient of determination (R2) of the first two ordination axes to each bioclimatic variable.**

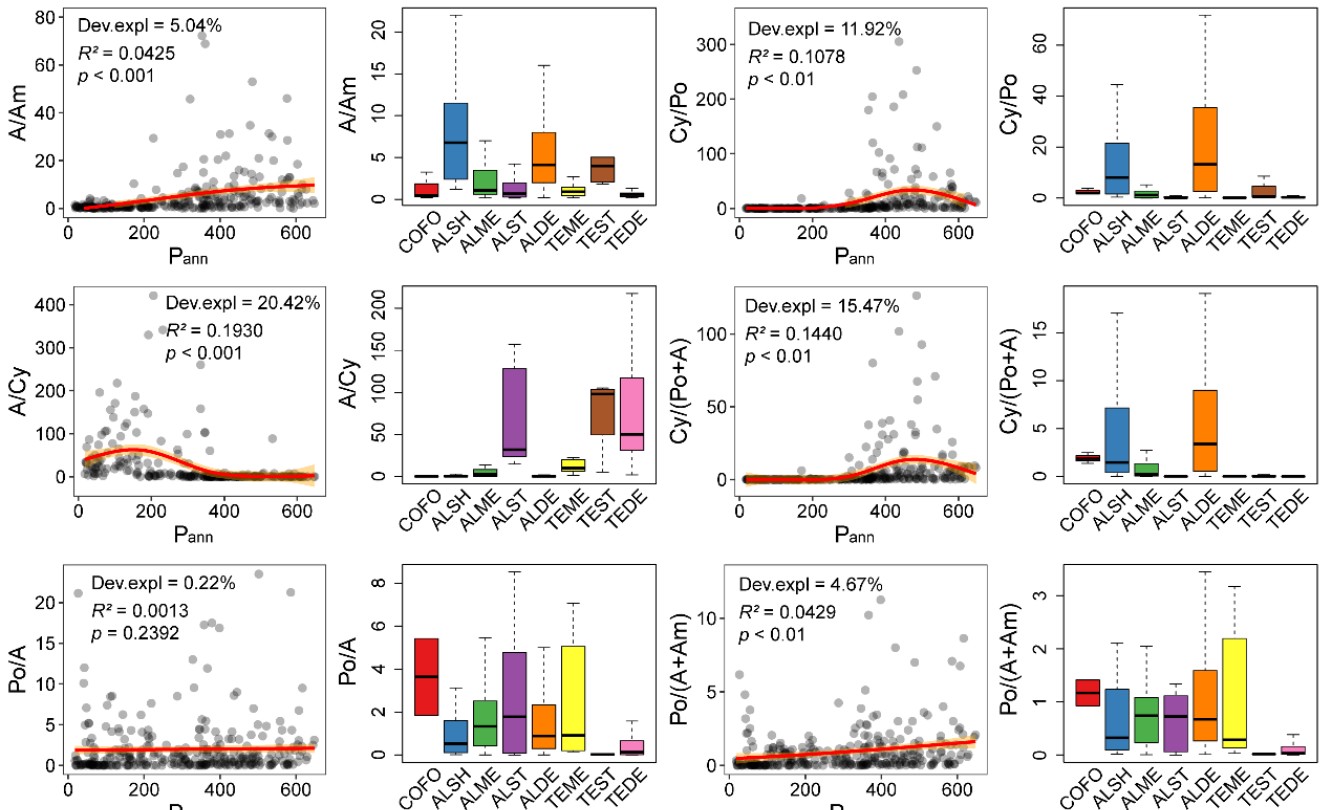

**Figure 6: Results of the generalized additive models (GAMs) illustrating the relationships between each pollen ratio and MAP (mm), and boxplots comparing each pollen ratio across different vegetation types. COFO: coniferous forest, ALSH: alpine shrub, ALME: alpine meadow, ALST: alpine steppe, ALDE: alpine desert, TEME: temperate meadow, TEST: temperate steppe, TEDE: temperate desert.**

From the distribution patterns of the main pollen taxa (Fig. 4) and the results of NMDS (Fig. 5), it is possible to distinguish vegetation types based on pollen assemblages. Especially, coniferous forest, alpine shrubland, and alpine meadow can be distinguished from other vegetation types (Fig. 5). Additionally, vegetation types with low precipitation or moisture requirements can probably be distinguished, as evidenced by the sites that score relatively high on the first axis (NMDS1) and relatively low on the second axis (NMDS2) (Fig. 5). Pollen assemblages of these sites are characterized by relatively high percentages of *Tamarix*, *Pinus*, Crassulaceae, Boraginaceae, and Poaceae (Fig. 4, Fig. 5). However, it is difficult to completely separate coniferous forest, alpine shrubland, alpine meadow, and temperate meadow, as well as distinguish temperature meadow, alpine steppe, temperature steppe, and alpine desert based on the pollen assemblages.

A current theme in pollen analysis research is the quantitative reconstruction of land cover using models of the relationships between pollen assemblages and the surrounding vegetation (Sugita, 2007a, b; Xu et al., 2016; Liu et al., 2023). Previous studies have demonstrated the potential of pollen assemblages to reflect dynamics of vegetation cover on the central and



eastern TP (Liu et al., 2023) and on the northeastern TP (Wang et al., 2023). The samples recorded in our dataset distribute across a wide spatial range, spanning from the southwest to the northeast of TP, and is accompanied by the published plot-

level vegetation data, which includes species identities, their respective numbers, and coverage for each species (Jin et al., 2022). Therefore, we believe that the combination of our dataset with the published plant community dataset will undoubtedly benefit relevant research conducted on a broader scale. Besides, Liu et al. (2023) proposed that pollen concentration is superior to pollen percentage for the quantitative reconstruction of vegetation cover. Our dataset includes the dry weight of each sample alongside the counted numbers of *Lycopodium* spores in each sample, allowing to estimate the

pollen concentration for each sample or site. Consequently, we believe that this dataset holds the potential to improve the accuracy of vegetation cover reconstruction on TP.

Pollen-assemblage diversity has been extensively applied to reconstruct variations in floristic diversity over time, although the reliability of the modern pollen-plant diversity relationship varies considerably among different regions (Meltsov et al., 2011; Goring et al., 2013; Felde et al., 2016; Reitalu et al., 2019; Connor et al., 2021; Cui et al., 2023; Liao et al., 2024). This

modern pollen dataset, coupled with its corresponding plant community dataset (Jin et al., 2022), has already been used to investigate the associations between pollen richness and evenness with plot-level plant richness and evenness, and with key climatic variables, landscape characteristics, and human disturbances (Liao et al., 2024). Further research can delve into other diversity metrics, such as the Shannon index (or Shannon-Wiener index), and β diversity indices, to comprehensively assess the reliability of the pollen-plant diversity relationship on TP.

In recent years, trait paleoecology, which couples modern data on plant functional traits to fossil pollen assemblages, has aroused widespread interest among palynologists and paleoecologists (Carvalho et al., 2019; Birks, 2020; Adeleye et al., 2023; Wang et al., 2024). Our dataset, along with corresponding datasets on plot-level functional traits (Jin et al., 2023) and plant communities (Jin et al., 2022), provide a valuable opportunity to assess the reliability of this approach in reflecting the long-term ecological properties of ecosystems on TP (Liao et al., in preparation). Phylogenetic diversity, another aspect of

biodiversity, has rarely been used in pollen analysis but has potential to enhance our understanding of long-term patterns of community assembly (Blaus et al., 2020). By leveraging published phylogeny data for tens of thousands of plant species (Zanne et al., 2014; Smith and Brown, 2018), it is possible to evaluate the relationships between phylogenetic diversity estimated from pollen and plant assemblages using our modern dataset and the corresponding plant community dataset. Such an assessment can further provide a foundation for applying this approach to elucidate community responses to long-term

changes in climate and human disturbance.

## 6 Data availability

The dataset includes pollen count data of each sample, as well as geographic coordinate, altitude, and local vegetation type for each sampling site. The dataset is openly accessible at https://doi.org/10.11888/Paleoenv.tpdc.302015 (Liao et al., 2025).

**7 Summary**

We present and analyze 555 pollen data, identified from topsoil collected within plant plots across a vast area of TP and along the southern margin of Xinjiang that borders the plateau. This dataset provides the count data for each pollen taxon in each sample, along with the location details (latitude, longitude, altitude) of each sampling site. Ordination analysis and multiple regression reveal that annual precipitation and precipitation of the wettest quarter are the primary factors influencing the spatial distribution of pollen assemblages across the entire study area. Additionally, ordination analysis

demonstrates that samples from coniferous forest, alpine shrubland, and alpine meadow can be distinguished from those from other vegetation types, and samples from vegetation types with low precipitation or moisture requirements from other samples, based on pollen assemblages in the dataset. Generalized additive models demonstrate that the six commonly used pollen ratios (A/Am, A/Cy, Po/A, Cy/Po, Cy/(Po+A), and Po/(A+Am)) are not sufficiently reliable in reflecting precipitation changes at a geographical scale across the entire study area. Nevertheless, they can provide some insight into changes in

vegetation or landscape. In addition to reconstruct changes in climate and vegetation, this modern pollen dataset can also be used to assess the reliability of pollen assemblages in representing dynamics of vegetation cover, functional traits, and plant diversity (from taxonomic diversity to functional and genetic diversity), by integrating corresponding datasets of plot-level plant communities and functional traits.

The modern pollen dataset present here covers most area of TP except the Hoh Xil uninhabited area in the northern

hinterland of the plateau. It fills a geographical gap in the published datasets that provide pollen count data for TP and its surrounding areas. In addition, our team has collected 314 topsoil samples from the Hengduan Mountains, a region characterized by its morphometric complexity and the largest elevation difference in the southeastern part of the plateau. We are currently in the process of identifying pollen from these samples and hope to soon incorporate them into the existing pollen datasets from TP.

**Author contributions**

ML, KL and JN designed the study. KL leaded the field trip. KL and YJ collected the samples. XC, LL and KL treated the samples and guided the pollen identification. ML analyzed the data and drafted the manuscript. KL, JN, YJ, LL and XC reviewed the manuscript.

**Competing interests**

The contact author has declared that none of the authors has any competing interests.



**Competing interests**

We thank Xinxin Zhou, Ang Liu, Linfeng Wei, Kai Wu, Jie Xia, Haoyan Wang, Dewu Xu, Luosang Tudan, Pingyu Sun, Yezi Sheng, Borui Zhou, Jing Hu, Ying Hou, Yang Yang, Hailu Zhong, and all the other students for their assistance with plant investigation and sample collection during the field works from 2018 to 2021, and with pretreatment in the labs. We appreciate Nannan Wang, Yaqin Hu, Yumei Li, and Linjing Liu for their help with pollen identification.

**Financial support**

This research was financially supported by the Second Tibetan Plateau Scientific Expedition and Research Program (Grant number: 2019QZKK0402) and the Strategic Priority Research Program of the Chinese Academy of Sciences (Grant number: XDA2009000003).

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
