# Peer review of "Modern Pollen Dataset of the Tibetan Plateau"

_Earth System Science Data, 2025_

## Author Comment (AC3)

[Figure]

The intensity of human disturbance on the Tibetan Plateau

[Figure]

Figure 5: Biplots of non-metric multidimensional scaling (NMDS) showing the sites (left) and the main pollen taxa (right). The inserted polar plot showing the coefficient of determination ($R^2$) of the first two ordination axes to each selected bioclimatic variables and intensity of human disturbance (HD). The upper panel displays the results from all sampling sites, while the lower panel shows the results excluding sites in coniferous forest, taxa from arboreal trees, and those with pollen counts of less than 200 grains. BIO1: Annual Mean Temperature, BIO2: Mean Diurnal Range, BIO3: Temperature Seasonality, BIO4: Isothermality, BIO12: Annual Precipitation, BIO14: Precipitation of Driest Month, BIO23: Moisture Index.

---

## Author Response (AR1)

**1# reviewer**

Overall, this paper only utilized simple ordination analysis and multiple regression analysis to briefly examine the indicative significance of surface pollen on vegetation types and climate in the Tibetan Plateau. It did not carefully analyze the relationship between pollen assemblages of different vegetation types and the actual vegetation. In the western and northern Tibetan Plateau, the vegetation is sparse, dominated by steppe-desert, but the surface pollen contains an uncertain amount of arboreal pollen, which is evidently transported from distant sources. The proportion and types of long-distance transported pollen may vary across different locations, and the sources could also be different - these are issues that need to be cautiously considered when reconstructing past vegetation and climate based on surface pollen (quantitative) data. Therefore, this paper only provides some surface pollen data, without clear implications for other disciplines.

**Reply:** As you may know that this paper is a data paper, the main purpose of this paper is to provide original pollen count data. We believe that these data will be beneficial for future researchers in conducting diverse studies related to vegetation/paleovegetation, climate change, and other related fields. Revolving around this purpose, we have provided comprehensive details on sample collection and pretreatment, methods for pollen identification, frequency and spatial distributions of pollen count for both samples and sites, the spatial distributions of the main pollen types, and the frequency distribution of the number of identified pollen types from different vegetation types. Through the application of statistical analyses, we have proposed potential uses for the dataset and cautionary notes for its utilization. Based on the results obtained, we aim to show the readers that whether or not the pollen data from this dataset can be used to distinguish vegetation types or can be reliably used to distinguish which vegetation types, and which climatic parameters are most promising to reconstruct based on the pollen data from this dataset. As for your comment that "It did not carefully analyze the relationship between pollen assemblages of different vegetation types and the actual vegetation.", we understand your concern that our pollen data processing and selection methods may not have adequately ensured representation of the actual vegetation types. However, regarding the question of how to effectively process the data to enhance the reliability of the pollen data in representing vegetation types or climatic parameters—such as excluding arboreal taxa from data derived from non-forested area, or using representation value or relative pollen production to adjust the percentages of specific pollen taxa—we consider this a potential research area of interest for future researchers who will utilize this dataset, potentially leading to the publication of further research papers.

For the second question, we completely agree with you that pollen data collected from open landscape contains an uncertain amount of long-distance transported pollen types, with potential source areas extending tens of kilometers beyond the sampling sites. Undoubtedly, this issue must be meticulously considered when reconstructing past vegetation and climate

using surface pollen data. We have emphasized this concern in the section "5 Potential usage of the dataset". Besides, we have added another result from the ordination analysis conducted on a processed pollen dataset that excludes samples from forested areas and removes arboreal pollen types. By doing so, we aim to show readers whether excluding arboreal pollen types from data collected in open landscape can enhance the indicative significance of the pollen data to vegetation types. Please refer the results to the updated Figure 5 in the revised manuscript.

Q1 (Line 10): This dataset fills a geographic gap in the published pollen count data for this region. To allow the reader to clearly see which locations are covered by this new dataset, the geographic distribution of the previous data should be indicated in Figure 1.

**Reply:** Thanks very much for this suggestion. We have revised Figure 1 to include the geographical locations of the original pollen count data from our study (indicated by circles) and those from other publicly available sources (indicated by triangles, including Cao et al., 2021 and Ma et al., 2024). Please refer to the updated Figure 1 shown below.

**Q2 (Lines 23-24): Italic**

**Reply:** Thanks for pointing out this problem. We have corrected the font of "Lycopodium" to italicized in the revised manuscript.

**Q3 (Line 73): The References for the Tibetan Plateau boundary.**

**Reply:** thanks for this suggestion: The reference related to the boundary of the Tibetan Plateau is: Zhang, Y., Li, B., Liu, L., and Zheng, D. (2021) Redetermine the region and boundaries of

Tibetan Plateau, Geographical Research, 40, 1543-1553, https://doi.org/10.11821/dlyj020210138, 2021. We have included this information in the caption of Figure 1 and added the corresponding reference to the reference list. Please refer to Lines 76 and 469-470.

Q4 (Line 79): In the transition zone around 3,000 meters of elevation, how can the delineation of distinct vegetation types be ensured

**Reply:** We did not attempt to distinguish different vegetation types based on altitude. Rather, we just want to state a fact that, the samples we collected from temperate vegetation generally distributed between 800-3000 m a.s.l. and those from alpine vegetation above 3000 m a.s.l.. The methodology for determine vegetation type at the sampling site has been detailed in our previously published work (see the reference below or Lines 376-378 in the revised manuscript): Jin, Y.-L., Wang, H.-Y., Wei, L.-F., Hou, Y., Hu, J., Wu, K., Xia, H.-J., Xia, J., Zhou, B.-R., Li, K. & Ni, J. (2022) A plot-based dataset of plant community on the Qingzang Plateau. Chinese Journal of Plant Ecology, 46, 846-854. http://doi.org/10.17521/cjpe.2022.0174

**Q5** (**Line 83**): Whether the pollen assemblage results obtained are representative of the different sampling densities in each vegetation area, which vary widely. If there are only three samples in a coniferous forest area, are they representative of the entire coniferous forest area?

**Reply:** Thanks very much for this question. We acknowledge that three samples can hardly show the entire picture of pollen assemblages from a vegetation covering a large geographic space. We have pointed out this problem in the section "5 Potential usage of the dataset" and emphasize the uncertainty may arise when using these data. As detailed in the revised manuscript (Lines 195-198), and reiterated below:

"It should be noted that the first two NMDS axis cannot differentiate between samples collected from forested and non-forested areas (Fig. 5). This limitation may stem from partially nested distributions of these vegetation types, long-distance dispersal of arboreal pollen taxa, and/or the limited sample size, which reduces the accuracy of the pollen composition in representing forests."

**Q6** (Line 88): What is the weight of each sample?**

**Reply:** The dry weights of the samples in our dataset range from 5.25 to 32.7 g. Approximately 77% (427 samples) of the total samples have dry weights exceeding 20 g, 14.8% (82 samples) weigh between 10 and 20 g, and the remaining 8.2% (46 samples) weigh between 5 and 10 g. The dry weights of the samples used for pollen extraction were determined by the soil properties, for example, the grain size and the biomass density surrounding the sampling sites. As mentioned in the manuscript (Line 23), the dry weight of each sample has been included alongside the pollen data in the dataset.

**Q7** (Line 110): Italic**

Reply: Thanks for pointing out this problem. We have made correction in the revised

manuscript.

**Q8** (Line 119): Does such a large gap in pollen grain counts all have an impact on analyses such as NMDS?**

**Reply:** Intuitively, we believe that a large gap in pollen counts would have some impact on the analyses, but the question on how significant the impact needs to be further studied. While the theoretical minimum threshold for reliable pollen identification is conventionally set at 300 grains, our discussions with researchers conducting alpine pollen studies on the Tibetan Plateau suggest that this benchmark may be adjusted to 200 grains in regions characterized by sparse vegetation cover and low species richness. A detail not sufficiently emphasized in the original manuscript is that the pollen data we used for analyses are the combined pollen data from multiple samples at each site. In the site-based dataset, a mere 9 pollen data have pollen counts fewer than 200 grains. In the revised manuscript, we reran the ordination analysis and regression models excluding these 9 data points and updated Figures 5 and 6 accordingly. We have added these details in the section "3.3 Numerical analyses". Please refer to Lines 102-103, 125-126 in the revised manuscript. The updated Figures 5 and 6 are shown below.

Figure 5: Biplots of non-metric multidimensional scaling (NMDS) showing the sites (left) and the main pollen taxa (right). The inserted polar plot showing the coefficient of determination (R2) of the first two ordination axes to each selected bioclimatic variables and intensity of human disturbance (HD). BIO1: Annual Mean Temperature, BIO2: Mean Diurnal Range, BIO3: Temperature Seasonality, BIO4: Isothermality, BIO12: Annual Precipitation, BIO14: Precipitation of Driest Month, BIO23: Moisture Index.

Figure 6: Results of the generalized additive models (GAMs) illustrating the relationships between each pollen ratio and MAP (mm), and boxplots comparing each pollen ratio across different vegetation types. COFO: coniferous forest, ALSH: alpine shrub, ALME: alpine meadow, ALST: alpine steppe, ALDE: alpine desert, TEME: temperate meadow, TEST: temperate steppe, TEDE: temperate desert.

**Q9** (Lines 122-125): Figure 2 shows that 31% of the samples contain between 200 and 300 pollen grains, and 44% contain between 400 and 600 grains. In other words, 75% of the samples have more than 200 grains. However, the key question is what percentage of samples contain fewer than 100 grains. While Figure 2 indicates the sample points with under 100 grains, it does not specify the overall percentage. If more than 20% of the samples have under 100 grains, the dataset should be reorganized to remove those low-grain samples.

**Reply:** In our dataset, there are 310 samples have pollen count more than 300 grains, which occupy 55.86% of the total samples. 168 samples (30.27%) have pollen count between 200-300 grains, 43 samples (7.75%) between 100-200 grains, and 34 samples (6.12%) have pollen count less than 100 grains. When considering pollen count for each site (as you may know from the manuscript that, we generally collected multiple samples from each site), there are only 3 sites (TP2019071102, TP2019071103, TP2019071202) have pollen count fewer than 100 grains. We retained these data in the dataset because we believe they might be useful to reflect some characteristics of the vegetation. In addition to pollen count data, our dataset includes supplementary data essential for calculating pollen concentration: (1) dry weight of each sample used for pollen extraction, (2) *Lycopodium* spore counts per tablet, and (3) identified *Lycopodium* spores per sample. Some researchers might be interested in testing the reliability of using pollen concentration to reflect the sparseness of vegetation or can use pollen concentration to explore other scientific questions. We realized from this question you raised

that it is necessary to provide more details on the pollen counts for samples and sites. So, we have revised the description about the pollen count in section 4.1. Please refer to Lines 133-138.

**Q10 (Line 161): Is it possible to consider the effect of altitude on pollen assemblages?**

**Reply:** If we understand you question correctly, you are asking whether the pollen assemblage from our dataset can reflect effectively vegetation changes along the altitude gradient. The answer is positive as evidenced by the pollen diagram shown below. However, it is important to note that the relationship between pollen assemblages and altitude is not "linear"; in other words, pollen assemblages vary across different altitude ranges. Within a specific altitude range, such as between 4065-4520 m, pollen assemblages exhibit limited sensitivity for detecting fine-scale elevational gradients.

Q11 (Line 173): Why does the A/Am also show high values in the alpine desert. The pollen assemblage should be dominated by Amaranthaceae.

**Reply:** As you can observe in Figure 4, Amaranthaceae comprises approximately 30~50% of the pollen taxa in alpine deserts, whereas *Artemisia* is even more abundant, reaching up to 50~70% or even higher. This explains why we obtain higher values of the A/Am ratio for this vegetation type. According to the plant survey data collected from the same sample sites, Amaranthaceae indeed dominates in temperate desert, but this is not the case in alpine deserts. For more detail, you may refer to the published plant community dataset in the Chinese Journal of Plant Ecology (<a href="https://www.plant-ecology.com/CN/10.17521/cjpe.2022.0174">https://www.plant-ecology.com/CN/10.17521/cjpe.2022.0174</a>). Besides, as you surely know that the representativeness of Amaranthaceae and *Artemisia* to their parent plants vary significantly across different vegetation types, which is another potential reason why the A/Am ratio is high in alpine deserts on the plateau.

**#Reviewer 2**

**General Comments**

1. Regarding the impact of climate on pollen assemblages: while the results at the plateau scale suggest precipitation is the most influential factor, considering the spatial heterogeneity of the Tibetan Plateau, would region-specific statistical analyses yield different outcomes? Additionally, only temperature and precipitation were considered as climate variables—could other climatic factors, such as solar radiation or wind speed, also play a role?

**Reply:** Thanks for your suggestions. We agree and admit that regional-specific statistical analyses must be yield different results. However, we must address that our work takes the plateau as whole. Therefore, we believe there is no need to conduct regional-specific analysis. Furthermore, we also agree that other climatic and topographic factors must cast considerable impacts on vegetation and pollen compositions. We have data for 23 bioclimatic variables. However, we consider it inappropriate to include all of these variables in the ordination analysis due to issues of collinearity among them and the possibility of resulting biplots being poorly readable and difficult to interpret. Therefore, we firstly conducted Pearson correlation tests between each pair of variables and eliminated those with a correlation coefficient greater than 0.8. Ultimately, seven bioclimatic variables were selected for the ordination analysis: BIO1 (Annual Mean Temperature), BIO2 (Mean Diurnal Range), BIO3 (Temperature Seasonality), BIO4 (Isothermality), BIO12 (Annual Precipitation), BIO14 (Precipitation of Driest Month), and BIO23 (Moisture Index). Additionally, we included the intensity of human disturbance (HD) as a predictor in the updated ordination analysis. HD shows low to moderate correlations (r

2. Based on local vegetation data, only a small proportion of the samples/sites are classified as forest, yet the abundance of forest taxa ranks second only to temperate desert taxa. This discrepancy may be influenced by pollen productivity and dispersal distance. A brief discussion on this aspect would be valuable.

**Reply:** Yes, only a few forest quadrats are investigated at eastern Tibetan Plateau, while the majority of the quadrats are located in the middle and western Tibetan Plateau, featuring vegetation types such as meadow, steppe, and desert. We admit that the majority of the pollen data contain arboreal pollen types, which must have been transported from regional forests within or even beyond the plateau. It is clear that this issue roots in pollen productivity and dispersal distance. We have added a brief discussion on this aspect in the section "4.1 Pollen count and taxa" (Lines 147-152), as presented below:

It should be noted that, due to the limited sample size (n = 3) from coniferous forest, the observed abundance of pollen taxa for this vegetation type may be subject to underestimation. Additionally, issues related to differences in pollen production and dispersal capacity may also influence the abundance of identified pollen taxa. Pollen types with a high pollen production and wide dispersal commonly dominate a pollen assemblage, thereby reducing the probability of detecting pollen types with a poorer representation and/or a low production in the pollen source area (Odgaard, 1999; Weng et al., 2006)."

3. The manuscript frequently compares the presented dataset with other published datasets. Beyond differences in data format (percentages vs. counts), please elaborate on any differences, such as differences in nomenclature and classification systems. If researchers aim to integrate your dataset with others, what considerations should be taken into account?

**Reply:** This is really important comments that will benefit us and other researchers. We have included detailed information on the nomenclature and classification systems used for pollen types in our dataset. Additionally, we propose specific considerations for merging this dataset with others (Lines 158-162), as outlined below:

"The nomenclature of pollen types in our dataset generally follows the conventions outlined in the Flora of China (https://www.iplant.cn/). However, we have retained the pollen type Chenopodiaceae despite its current taxonomic replacement within the family Amaranthaceae as per the latest Angiosperm Phylogeny Group (APG) classification. Therefore, if researchers intend to integrate our dataset with others, they should pay particular attention to standardizing the nomenclature of pollen types."

**Specific Comments**

1. Figure 1: For visual clarity, please remove the minutes and seconds from the latitude and longitude labels.

**Reply:** We have made revision. Please refer to the updated Figure 1 presented below.

Line 105: Given the use of 1-km resolution climate data, does this resolution accurately represent the spatial scale inferred by modern topsoil pollen data? Please clarify the spatial representativeness of modern pollen assemblages.

Reply: To be frank, we are unable to clarify this issue due to vast spatial scale covered by our dataset. Atmospheric convection regimes on the plateau and its surrounding area are complex, with different atmospheric circulations prevail in different regions. Consequently, the dispersibility of a specific taxon may vary noticeably across variation parts of the study area. We admit that, in open landscapes, the pollen source area can be extensive, and the 1-km resolution employed in this study may not accurately represent the spatial scale inferred from modern topsoil pollen data. Given that our statistical analyses focused on the entire Tibetan Plateau and the southern margin of Xinjiang adjacent to the plateau, this expansive area displays pronounced environmental gradients. Consequently, we believe that 1-km climate data can adequately capture the overall climate gradient, as demonstrated by the Mean Annual Precipitation (MAP) gradient illustrated in Figure 4, and thereby yields reliable results.

2. The manuscript repeatedly refers to human disturbance. How do the levels of human impact vary across different sites in this dataset? Has human activity altered the composition of pollen assemblages?

**Reply:** According to the published plot-level vegetation data (Jin et al., 2022, http://doi.org/10.17521/cjpe.2022.0174), the intensity of human disturbance was classified into to 3 levels (1 to 3), representing "weak", "moderate", and "strong", respectively. From the figure below, you can find that strong human disturbance is concentrated along the southern margin of Xinjiang adjacent to the plateau and in the northeastern part of the Tibetan Plateau. In contrast, weak human disturbance is predominantly observed in the southwestern part of the plateau, while moderate disturbance is primarily distributed in the central or south-central region of the plateau.

To figure out whether human activity has altered the composition of pollen assemblages in our dataset, we have included human disturbance as a predictor variable when conducting the ordination analysis. The result indicates that human disturbance did have some impact on the composition of pollen assemblages, though this impact was not crucial. Please refer to the updated Figure 5 in the revised manuscript.

3. Line 215: The integration of pollen data with plant functional traits offers deeper insights into long-term vegetation responses to past climatic disturbances. Please briefly explain how data of differing resolutions were integrated in this analysis.

**Reply:** If we understand corerctly, you are referring to discrepancies in spatial resolution between the pollen source area and the size of the plant quadrat. We acknowledge that the pollen source area in open vegetation is significantly larger than the spatial scale of the vegetation surveys (1 m×1 m for meadow, steppe, and desert, and 2 m×2 m or 5 m×5 m for shrubland). The distinct difference in spatial scale between the pollen source area and the vegetation survey may result in a mismatch between the relative abundance of pollen taxa and the relative importance of their parent plants. Consequently, this discrepancy will inevitably introduce bias when calculating indices of plant functional traits using pollen data. We have mentioned this issue in another paper comparing functional traits derived from pollen and plant assemblages, which is currently under review. Frankly, resolving such a discrepancy remains challenging.

4. Line 246: The dataset's continued expansion is anticipated, particularly in the southeastern and northwestern regions of the Tibetan Plateau.

**Reply:** Our team has collected over 300 topsoil samples from the Hengduan Mountains, which encompass some areas in the southeast of the Tibetan Plateau. However, data collection remains challenging in the uninhabited regions located in the northwest of the plateau.

---

## Author Response (AR2)

**Topic editor:**

The revised manuscript has satisfactorily addressed the referees' comments and is accepted for publication in ESSD. Please ensure that the formatting, figures, and data deposition fully comply with the ESSD requirements.

**Reply:** We appreciate for the editor's positive comments on our paper. We confirm that the final version of our manuscript complies with ESSD's requirements regarding formatting, figures, and data deposition. Additionally, we have carefully reviewed the text and corrected grammatical errors and expression issues.